

# Improving heat transfer predictions in heterogeneous riparian zones using transfer learning techniques

Aohan Jin[1], Wenguang Shi[1], Renjie Zhou[2], Hongbin Zhan[3], Quanrong Wang[1, 4*] and Xuan Gu[1]

[1]School of Environmental Studies, China University of Geosciences, Wuhan, Hubei, 430074, PR China.

[2]Department of Environmental and Geosciences, Sam Houston State University, Huntsville, TX 77340, USA.

[3]Department of Geology and Geophysics, Texas A&M University, College Station, TX 77843-3115, USA.

[4]MOE Key Laboratory of Groundwater Quality and Health, China University of Geosciences, Wuhan 430078, PR China.

*Correspondence to*: Quanrong Wang (wangqr@cug.edu.cn)

## Abstract.

Data-driven deep learning models usually perform well in terms of improving computational efficiency for predicting heat transfer processes in heterogeneous riparian zones. However, traditional deep learning models often suffer from accuracy when

data availability is limited. In this study, a novel deep transfer learning (DTL) approach is proposed to improve the accuracy of spatiotemporal temperature distribution predictions. The proposed DTL model integrates the physical mechanisms described by an analytical model into the standard Deep Neural Networks (DNN) model using a transfer learning technique. To test the robustness of the proposed DTL model, the influence of the number of observation points at different locations, streambed heterogeneity ($\sigma_{lnK}^2$ =0, 0.2, 0.5, and 1.0), and observation noise levels ($\sigma_{Noise}$ =0.025, 0.05, 0.075) on the MSE values

between the observed and predicted temperature fields. Results indicate that the DTL model significantly outperforms the DNN model in scenarios with scarce training data, and the mean MSE values decrease with increasing observation points for both DTL and DNN models. The mean MSE values for both the DTL and DNN models approach zero as the number of observation points increases to 200, indicating that both DTL and DNN models perform satisfactorily. Furthermore, increasing $\sigma_{lnK}^2$ and $\sigma_{Noise}$ raises the mean MSE values of the DTL and DNN models, with the DTL model exhibiting greater robustness

than the DNN model, highlighting its potential for practical applications in riparian zone management.

**Keywords**: Heat transfer, Riparian zones, Heterogeneity, Transfer learning



# 1 Introduction

Understanding heat transfer processes in riparian zones is critical for evaluating physical and biochemical processes during surface water-groundwater interactions, such as contaminants transport (Elliott and Brooks, 1997; Schmidt et al., 2011), water resources management (Bukaveckas, 2007; Fleckenstein et al., 2010) and aquatic ecosystems regulation (Ren et al., 2018; Halloran et al., 2016). As a primary source of uncertainty in riparian zone modeling, the inherent heterogeneity of the streambed stands out as a pivotal factor in accurately modeling groundwater flow and heat transfer processes (Karan et al., 2014; Brunner et al., 2017). However, given the intricacies of streambed heterogeneity, data acquisition in heterogeneous riparian zone is often time-consuming and costly (Zhang et al., 2023; Kalbus et al., 2006). Consequently, achieving accurate predictions of heat transfer processes in heterogeneous riparian zones with limited observation data remains challenging.

Over the past few decades, there has been a substantial increase in efforts toward simulating heat transfer processes in riparian zones, which can be categorized into two groups: physics-based models and data-driven models (Barclay et al., 2023; Feigl et al., 2021; Heavilin and Neilson, 2012). Typically, the physics-based models employ partial differential equations to characterize heat transfer dynamics within riparian zones, like the convection-diffusion equation (Chen et al., 2018; Keery et al., 2007), which aims to simulate and forecast temperature variations within riparian zones. Resolving the convection-diffusion equation generally involves two approaches: analytical and numerical models. Analytical models provide a precise mathematical representation of heat transfer dynamics and offer fundamental insights into physical processes within riparian zones, but their applicability is often limited to rather simplified and idealized scenarios (Keery et al., 2007; Bandai and Ghezzehei, 2021). Numerical models, which rely on





discretizing governing equations and solving them iteratively, are able to handle more intricate scenarios

and address unsteady flows effectively (Cui and Zhu, 2018; Ren et al., 2019; Ren et al., 2023).

Nevertheless, numerical models are constrained by the uncertainties of model structures and the

prerequisites of streambed characteristic parameters (Heavilin and Neilson, 2012; Shi et al., 2023).

Data-driven models, unlike physics-based models, can create a direct mapping between input and output

variables without explicit knowledge of underlying physical processes governing the system (Zhou and

Zhang, 2023; Callaham et al., 2021). In recent years, data-driven models have achieved significant

advancements and emerged as a successful alternative in hydrological and environmental modeling (Zhou

et al., 2024; Cao et al., 2022; Wade et al., 2023). However, their deficiency in incorporating physical

principles restricts their capability to delineate explicit computational processes as physics-based models,

posing a challenge to achieve enhanced extrapolation capabilities (Read et al., 2019; Cho and Kim, 2022).

Meanwhile, data-driven models typically require massive amounts of data for training and may yield

results that defy established physical laws due to the lack of physical principles (Read et al., 2019; Xie et

al., 2022). The strengths and weaknesses inherent in both data-driven and physically-based models are

evident across various research domains (Kim et al., 2021; Wang et al., 2023). Consequently, there is an

increasing inclination towards integrating physical processes into data-driven models, which enables

these models to extract patterns and laws from both observation data and underlying physical principles

(Zhao et al., 2021; Karpatne et al., 2017).

Transfer learning provides a feasible approach for integrating analytical and DL models, where

knowledge is transferred from a distinct but relevant source domain to enhance the efficacy of the target

domain (Zhang et al., 2023; Chen et al., 2021). This approach can diminish the requirement for extensive

training data in the target domain, which is considered as a major barrier of DL applications. By

leveraging knowledge gained from pre-training models, it accelerates the learning process and enhances

model performance (Guo et al., 2023; Jiang and Durlofsky, 2023). Recently, the use of the transfer

learning technique has gained attention in the field of hydrological modeling (Zhang et al., 2023; Cao et

al., 2022; Chen et al., 2021; Vandaele et al., 2021; Willard et al., 2021). For example, Xiong et al. (2022)

developed an Long-short term memory (LSTM) model of daily dissolved inorganic nitrogen

concentrations and fluxes in the coastal watershed located in southeastern China. They retrained this

model using multi-watershed data and successfully applied it to seven diverse watersheds through transfer

learning approach. Zhang et al. (2023) used the transfer learning technique to integrate the deep learning

model and analytical models for predicting groundwater flow in aquifers and obtained satisfactory

prediction performance for complex scenarios.

In this study, we introduce a novel deep transfer learning (DTL) approach that incorporates physical

information from analytical models into a deep learning framework using the transfer learning technique.

The proposed DTL model is implemented to predict the spatiotemporal temperature distribution in

heterogeneous riparian zones by leveraging analytical solutions, deep learning models, and transfer

learning. The analytical model is used to efficiently produce physically consistent heat distribution

patterns and data in homogeneous riparian zones, which serve as the training data for the pre-training

deep learning model. Subsequently, the weights and biases learned from the pre-training model are

transferred to a new deep learning model under heterogeneous scenarios through transfer learning. By

integrating insights from analytical models with the approximation power of deep learning models, the

DTL model achieves improved efficiency and performance. Notably, the newly proposed demonstrates

significant performance improvement, even with scarce observational data. This innovative approach

provides for accurate and efficient modeling of complex heat transfer processes in heterogeneous

environments, even with limited observation data.

## 2 Methods

### 2.1 Conceptual model

The two-dimensional (2D) conceptual model of the heat transfer process in a heterogeneous streambed is

depicted in Figure 1. The coordinate system originates at the center of the river, with the x-axis orientated

horizontally from left to right along the streambed. The $z$-axis is located vertically downward along the

left inlet boundary of the system and perpendicular to the $x$-axis. It is postulated that the thermal and

hydraulic properties of the streambed maintain uniformity. The river has a width of $2L$. Heat originated

from the river, with its temperature represented by an arbitrary function. The initial and boundary

conditions are depicted in Figure 1. An initial temperature of 20 °C is prescribed. The boundary conditions

on the left, right, and bottom sides are all specified as no heat flow boundaries. The top boundary condition

at $0 \le x \le L$ ($L = 0.32\ m$ in this study) is represented by a sinusoidal temperature signal ranging

between 19 and 21°C (i.e., $f(t) = 20 + sin(2\pi t)\ °C$). Meanwhile, the top boundary condition at $x > L$

is held constant at a temperature of 20 °C. The initial and boundary conditions are adopted from Shi et al.

(2023). The details of the analytical solution for the homogeneous streambed and the numerical solution

for the heterogeneous streambed are available in the Supplement.



## 2.2 Deep neural network (DNN)

The deep neural network (DNN) is a multi-layer feed-forward network with an input layer, multiple

hidden layers, and an output layer. The backpropagation algorithm is utilized to minimize the mean error

of the output and has been proven to be crucial in enhancing convergence (Jin et al., 2024). Assuming the

presence of $m$ hidden layers, the input and output vectors are denoted by $X$ and $O$, respectively. The

forward equations of the DNN model can be represented as follows:

$$H_1 = tanh(W_1 X + b_1) \tag{1a}$$

$$H_2 = tanh(W_2 X + b_2) \tag{1b}$$

$$H_m = tanh(W_m X + b_m) \tag{1c}$$

$$O = tanh(W_{m+1} X + b_{m+1}) \tag{1d}$$

where $H_i$ represents the output of the $i$-th hidden layer; $W$ and $b$ represent the weight matrices and bias

vectors, respectively. Typically, $W$ and $b$ can amalgamate as the parameter set $\theta = \{W_i, b_i\}_{i=1}^{m+1}$, $tanh$

refers the $tanh$ activation function. These parameters can be estimated by minimizing the following loss

function:

$$\theta = \underset{\theta^*}{argmin} \frac{1}{n} \sum_{i=1}^{n} |NN(X, \theta) - y_i|^2 \tag{2}$$

To mitigate the impact of dimensionality during the training process, the temperature field dataset is

normalized to $[-1,1]$ through the following equation in the pre-training process:

$$D_{norm} = 2 \frac{D - D_{min}}{D_{max} - D_{min}} - 1 \tag{3}$$

where $D$ denotes data utilized in the DNN model, $D_{max}$ and $D_{min}$ denote the maximum and minimum values and are computed with reference to the training samples exclusively. The same values of $D_{max}$ and $D_{min}$ are employed to normalize the testing samples to prevent data leakage (Zuo et al., 2020).

## 2.3 Transfer learning

As depicted in Figure 2, traditional deep learning models are allocated to distinct learning tasks, which require each model to be trained independently from scratch, leading to high computational demands and the need for substantial amounts of training data for each task. In contrast, the transfer learning technique offers an efficient alternative. By employing a pre-trained model that is then fine-tuned for predicting heat transfer in heterogeneous streambeds, transfer learning can significantly reduce the computational

burden and the need for large datasets. This is achieved by leveraging the knowledge gained from the source domain and applying it to the target domain, thereby accelerating the learning process and improving the model performance.

The transfer learning technique involves training a model to establish a mapping between the input vector $X$ and the observed data $O$ derived from a target dataset $D = \{(x_i, o_i)_{i=1}^n, x_i \in X, o_i \in O\}$. It assumes that

both the source and the target tasks share similar parameters or prior distributions of the hyperparameters. The pre-training model is established utilizing the dataset from the source tasks $D_s = \{(x_s, o_s)_{s=1}^n, x_s \in X, o_s \in O\}$ which is generated through analytical or numerical models. In this study, the source and target datasets are spatiotemporal distributions of temperature fields in homogeneous and heterogeneous streambeds, respectively. The hyperparameters $\theta_T$ for the fine-tuning model is acquired through the

optimization of the loss function delineated by:



$$\theta_T = \underset{\theta_T}{argmin} \sum_{i=1}^{n^*} \frac{1}{n^*} |f(x_t; \theta_T) - o_t|^2 \tag{4}$$

where $n^*$ denotes the number of training datasets employed to fine tune the pre-training model, $f()$ denotes the predictive function of the fine-tuning model.

The flowchart of the newly proposed framework is summarized in Figure 3: the DTL model is developed by initially generating an input dataset using the analytical model for heat transfer in homogeneous streambeds. The dataset is subsequently employed to pre-train a DNN model, focusing on learning the weights and biases of the fully connected layer. Next, the data of the observation points in the corresponding numerical model for heat transfer in heterogeneous streambeds is utilized to fine-tune the pre-trained DNN model by transferring the learned insensitive layers (i.e., freezing their weights and biases) and retraining the learnable parameters of the remaining layers. Finally, the effectiveness of the DTL model is evaluated by comparing its performance against a traditional DNN model with different amount of observation points, which evaluates the model's ability to predict the spatiotemporal temperature distribution in heterogeneous streambeds.

## 3 Results

### 3.1 Pre-training process

In this study, the pre-training model is a DNN model with 6 hidden layers, each containing 16 neurons. To evaluate the sensitivity of weights and biases to hydraulic conductivity and to identify which layers should be trainable or remain frozen, two pre-training models with identical structures but varying hydraulic conductivities are constructed. Both datasets consist of a $100 \times 100$ grid with 100-time steps





generated by the 2D analytical model, where 80% of the dataset is utilized for training and the remaining

20% for testing. $q_x$ and $q_z$ depend on hydraulic conductivities in $x$ and $z$ directions. In this section, $q_x$

and $q_z$ are set to $0.2m/d$, $0.3m/d$ and $0.6m/d$, $0.9m/d$ for these two models, respectively. Results

indicate that the predictions of the two pre-training models closely align with the analytical model with

average MSE values of $1.2E-6$ and $1.5E-6$, respectively. Similar to the works of Hu et al. (2020) and

Zhang et al. (2023), the difference in weights and biases between the two pre-training models is evaluated

using the relative change rate (RCR):

$$RCR = \frac{1}{I}\sum_i^I \frac{|\theta_{1i}-\theta_{2i}|}{\theta_{1i}} \tag{5}$$

where $\theta_{1i}$ and $\theta_{2i}$ are parameter matrixes in two pre-training models, respectively, $I$ is the number of

elements in the parameter matrix. For enhanced comparability and credibility, each of the two pre-training

models undergoes 20 training processes. Figure 4 presents the average RCR of weights and biases across

all layers for two per-training models over 20 trials. The RCR of biases shows consistent stability across

all layers, except for layer 3. In contrast, variations in weights are more prominent, particularly in layers

1, 2, and 3, which underscores the heightened sensitivity of these layers to hydraulic conductivity.

Consequently, layers 1, 2, and 3 of the pre-training models are marked as trainable, while the remaining

layers are held frozen in the following analysis. Notably, the convergence criteria are defined as a

threshold of 3000 iterations with a minimum gradient alteration of $5E-6$ throughout the training phase.

## 3.2 Spatial and temporal performance for homogeneous scenario

The spatiotemporal distribution of temperature in homogeneous streambeds is obtained by the analytical

model. In this study, we use 1, 5, 10, 20, 50, and 100 observation points, each with 100-time steps. The





hydrological parameters are set at $q_x = 0.4 \, m/d$ and $q_z = 0.6 \, m/d$, with all other parameters being

consistent with those presented in Figure 1. The temperature data from these observation points are used

as training data for DTL and DNN, respectively. The reference temperature field on $0.5d$ (i.e., the $50th$

time step) is employed as testing data. Figure 5 illustrates the absolute errors of the DTL and DNN models

for the homogeneous riparian zone. The results suggest that the DTL model aligns well with the reference

temperature field, whereas the DNN model tends to struggle in accurately capturing the reference

temperature field. This highlights the significant improvement in the performance of the DL model

facilitated by prior knowledge of the analytical solution and physical information. The pre-training model

incorporates physical knowledge to provide superior initial parameters (weights and biases), which

narrows the search space during the fine-tuning process. In contrast, the DNN model randomly initializes

these parameters and requires more training points to explore the entire parameter space. To further

demonstrate the predictive performance of the proposed model in time series, Figure 6 shows the

temperature time series predicted by the DTL and DNN models at a given observation point ($x = 0.5m$,

$y = 0.5m$). Results indicate that the DTL model predicts the temperature fluctuation trend better

compared to the DNN model. Especially for the sparse dataset with a few observation points, the average

$MSE$ of the DTL model with 5 observation points is approximately 3.2 times lower than that of the DNN

model. As shown in Figure S2 in the Supplement, there is no significant difference in the performance of

the DTL and DNN models when the number of observations point increases to 200. Notably, the

performance of the DTL model appears to be less sensitive to the amount of observation points. We

attribute this phenomenon to two factors: (1) randomly selected observation points lead to optimal

performance when the observation points are in proximity to the test point, and vice versa; (2) the DTL

model demonstrates the capacity to integrate substantial information from the analytical model, which diminishes the requirement for the number of observation points.

The choice of observation points can influence the outcomes of the proposed DTL model. To mitigate the effect of their positions, each observation point is randomly generated 200 times. The distributions of the average *MSE* for both the DTL and the DNN models across diverse amount of observation points are illustrated in Figure 7. Results reveal that both the interquartile range and mean values of MSE for the DTL model are considerably smaller than those of the DNN model. As an illustration, when considering 10 observation points, the average MSE for the DTL model is approximately 0.12, whereas that for the DNN model is 0.54. Furthermore, there is a significant reduction in both interquartile range and mean values of MSE of the DTL model, and the interquartile range and mean values of MSE of the DTL model tend to stable as the amount of observation points exceeds 50. On the contrary, the interquartile range and mean values of *MSE* of the DNN model consistently decrease with an increasing amount of observation points, displaying a consistent pattern as observed in Figure 6. It should be emphasized that the DTL model can still produce satisfactory results even with sparse data. Even with more than 50 observation points, the DNN model still underperforms the DTL model, which can be attributed to the following reasons: (1) due to the lack of prior physical knowledge, the DNN model may require more data to learn relatively complex patterns; (2) both the DTL and the DNN model follow the identical convergence criterion with a restricted number of epochs during the fine-tuning process, which may result in incomplete training for the DNN model.



### 3.3 Effects of nonuniform flow on heat transfer

In this section, we evaluate the performance of the DTL model in predicting the spatiotemporal temperature distribution in heterogeneous streambeds. The heterogeneous $lnK$ field is generated by the exponential covariance function with mean $\mu = 0$, correlation length $l = 0.1m$ in both the $x$ and $z$ directions, variance $\sigma_{lnK}^2 = 0.2, 0.5$ and $1.0$, respectively. Accordingly, three scenarios with low to high heterogeneity are created. Figure 8 depicts the random $lnK$ fields and references flow fields of three scenarios. The other parameters remain consistent with those of the homogeneous streambed. The temperature distribution in the heterogeneous streambed is estimated using the numerical model. Temperature time series of 1, 5, 10, 20, 50 and 100 observation points are extracted to fine tune both the DTL and DNN models.

To mitigate the impacts of random sampling during the fine-tuning process, 200 stochastic simulations are performed. The distribution of the average MSE for both the DTL and DNN models in three distinct heterogeneous streambeds from low to high heterogeneity are shown in Figure 9. One can find that the average MSE of the DTL model is consistently minimal and significantly lower than that of the DNN model. Besides, with the same number of observation points, a decrease in $\sigma_{lnK}^2$ corresponds to a reduction in average MSE. These findings can be explained by the fact that the proposed DTL model exhibits a strong ability to transfer knowledge between two datasets with similar structures or features. A decreased $\sigma_{lnK}^2$ indicates less heterogeneity in the $lnK$ field, resulting in a temperature field that more closely resembles those generated by the analytical model. We attribute this improvement in the DTL model to the enhanced initial parameters of the DNN model through the incorporation of physical knowledge during the fine-tuning process. For both the DTL and DNN models, the interquartile ranges

and mean values of MSE decrease as the amount of observation points increases. Notably, by leveraging the insights from the analytical model, the DTL model can effectively predict the temperature distribution in heterogeneous streambeds, even with sparse observation points (e.g., 5 observation points). In contrast, while the DNN model exhibits improved performance with an increased amount of observation points,

its performance heavily relies on this factor, showing unsatisfactory outcomes with fewer observation points. When the amount of observation points reaches 50, the interquartile range and mean MSE of the DTL model exhibit marginal changes, but the interquartile range and mean MSE of the DNN model still decrease significantly. Furthermore, there is no significant difference in the performance of the DTL and DNN models in heterogeneous scenarios when the number of observation points increases to 200, as

shown in Figures S3 and S4 in the Supplement. The average MSE of the DNN model is approximately 2.8 to 18.4 times smaller than that of the DTL model with the same observation points, which further demonstrates the capability of the DTL model to transfer knowledge from homogeneous environments in heterogeneous environments.

### 3.4 Effects of river temperature uncertainty

In this section, we evaluate the effectiveness of the DTL model in the context of river temperature observation noises, which may arise from suboptimal field conditions or sensor resolution limitations (Chen et al., 2022; Shi et al., 2023). Specifically, the white Gaussian noise is introduced at the top boundary:

$$f(t) = 20 + sin(2\pi t) + normrnd(\varphi_{Noise}, \sigma_{Noise}) \tag{6}$$



where $\varphi_{Noise}$ and $\sigma_{Noise}$ denote the mean and variance of white Gaussian noise, respectively, and

$normrnd()$ denotes the Gaussian distribution. In this section, $\varphi_{Noise}$ is set to 0°C and $\sigma_{Noise}$ is set to

0.025°C, 0.05°C and 0.075°C, respectively, as shown in Figure 10. Similarly, the heterogeneous $lnK$ field

of streambed is generated by the exponential covariance function with $\mu = 0$, $l = 0.1m$ in both $x$ and $z$

directions and $\sigma_{lnK}^2 = 0.5$. The temperature time series from diverse numbers of observation points (1, 5,

10, 20, 50, and 100) are utilized as training datasets for both DTL and DNN models. Additionally, 200

stochastic simulations are conducted to mitigate the influence of random sampling of observation points

during the fine-tuning process.

Figure 11 shows the distributions of the average $MSE$ for both the DTL and DNN models under different

noise levels. It is observed that that the DTL and DNN models exhibit sensitivity to noise, and the elevated

noise levels result in diminished model performance. Nevertheless, the DTL model is less impacted by

river temperature uncertainty compared to the DNN model. For instance, in cases of 10 observation points,

the average MSE of the DNN model varies from 0.59 to 0.45 as $\sigma$ decreases from 0.075 to 0.025. In

contrast, the average MSE of the DTL model ranges only from 0.12 to 0.09 under the same conditions,

demonstrating the superior robustness of the DTL model over the DNN model.

**4 Discussions**

This study investigates the effects of streambed heterogeneity, temperature observation noises and the

number of observation points at different locations on the performance of the proposed DTL model.

Results indicate that the proposed transfer learning model exhibits robust prediction performance with

significantly reduced interquartile range and mean MSE, particularly in scenarios with sparse data. These

findings suggest that integrating analytical knowledge enables effective decrease of model uncertainties. It is worthwhile to point out that the framework developed in this study is not limited to heat transfer in riparian zones: it can also be applied to mass transport and heat transfer in other heterogeneous porous media. This versatility highlights the framework's potential for broader applications across various fields within environmental and hydrological studies. Future research will systematically explore the difference

between transfer learning-based models and conventional models for modeling heat transfer under uncertain conditions. However, it is imperative to recognize several constraints associated with the DTL model proposed in this study. Firstly, the incapacity for extrapolation of the DTL model restricts its applicability. As it lacks observation points outside the training domain, the DTL model tends to face limitations concerning extrapolative tasks. Secondly, this study centres on modeling heat transfer

problems in heterogeneous riparian zones, and the effectiveness of the DTL model may be influenced by the selection of the $K$ value. Finally, analytical models usually require regular spatial domains, while real-world study areas (e.g., watersheds) often feature irregular spatial domains. The effectiveness of the DTL model may be influenced by discrepancies between the temperature field in the real-world area and the simplified analytical solution, especially near the boundary. All these issues should be investigated

separately in the future.

## 5. Conclusions

In this study, we propose a novel deep transfer learning (DTL) approach, which enhances DNN models by integrating physical mechanisms described by an analytical model using transfer learning technique. The proposed DTL model is tested against the DNN model under different heterogeneous streambeds and

observation noise levels. Results indicate that the DTL model significantly improves the robustness and accuracy in predicting the spatiotemporal temperature distribution in heterogeneous streambeds by incorporating knowledge transferred from pre-trained DNN models. Importantly, the DTL model maintains satisfactory performance even with sparse training data and high uncertainties in geological conditions and observations, making it a promising tool for practical applications in riparian zone

management. This is particularly relevant in situations where data acquisition is often challenging and costly, highlighting the potential impact of our research. The main conclusions are summarized as follows:

(1) The hydraulic conductivity primarily influences the parameters of the shallow layers in the DNN model, rendering it visible to use transfer learning approach in predicting spatiotemporal temperature distribution in heterogeneous streambeds;

(2) The accuracy of predicted temperature fields for both the DTL and DNN models improves with an increased number of observation points, and the DTL model significantly outperforms the DNN model for both homogeneous and heterogeneous scenarios;

(3) The DTL model demonstrates stronger robustness in dealing with observation noise compared to the DNN model and performs satisfactorily even with sparse training data;

(4) The successful application of the DTL model for predicting the spatiotemporal temperature distribution in heterogeneous streambeds indicates its pronounced advantages and prospects for estimating surface water and groundwater interaction fluxes in such heterogeneous riparian zones.



**Data availability**

The Python codes of the DTL and DNN models are made available for download from a public repository

at: https://github.com/Ahjin-CUG/TL.

**Author contributions**

Aohan Jin: Methodology, Software, Visualization, Writing - original draft; Wenguang Shi: Methodology,

Validation, Writing - review & editing; Renjie Zhou: Conceptualization, Validation, Writing - review &

editing; Hongbin Zhan: Validation, Writing - review & editing; Quanrong Wang: Conceptualization,

Writing - review & editing, Funding acquisition, Project administration; Xuan Gu: Writing - review &

editing.

**Competing interests:**

The contact author has declared that neither they nor their co-authors have any competing interests.

**Acknowledgments**

This research was partially supported by Programs of the National Natural Science Foundation of China

(Grant 42222704).

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



**Figure Captions**

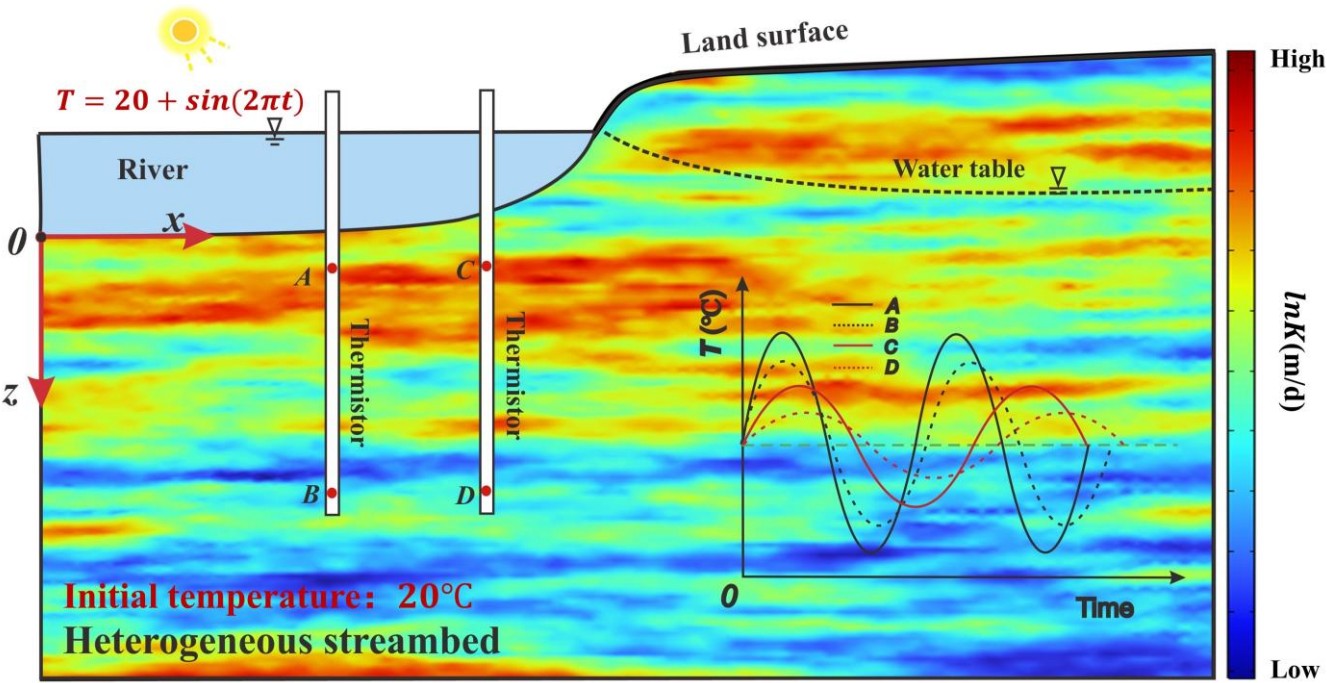

**Figure 1.** Schematic diagram of the temperature distribution in the riparian zones.



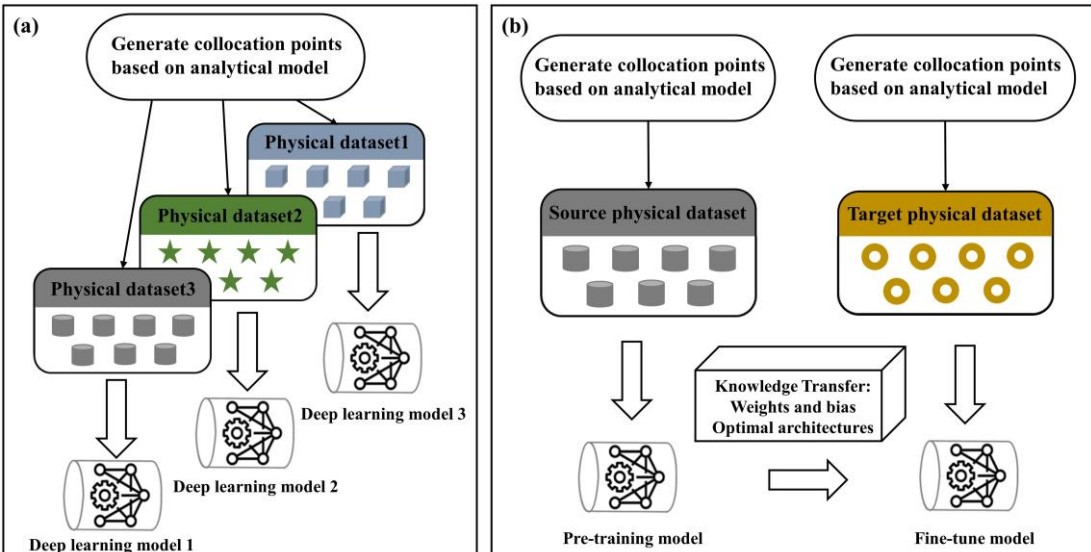

**Figure 2.** Schematic diagram of the pre-training and fine-tuning methods in the transfer learning model (Revised from (Guo et al., 2023)). (a) Traditional machine learning method; (b) Transfer learning method.



**Figure 3.** Proposed DTL framework used in this study. The framework consists of a pre-training module, a transfer learning module, and an evaluation module.



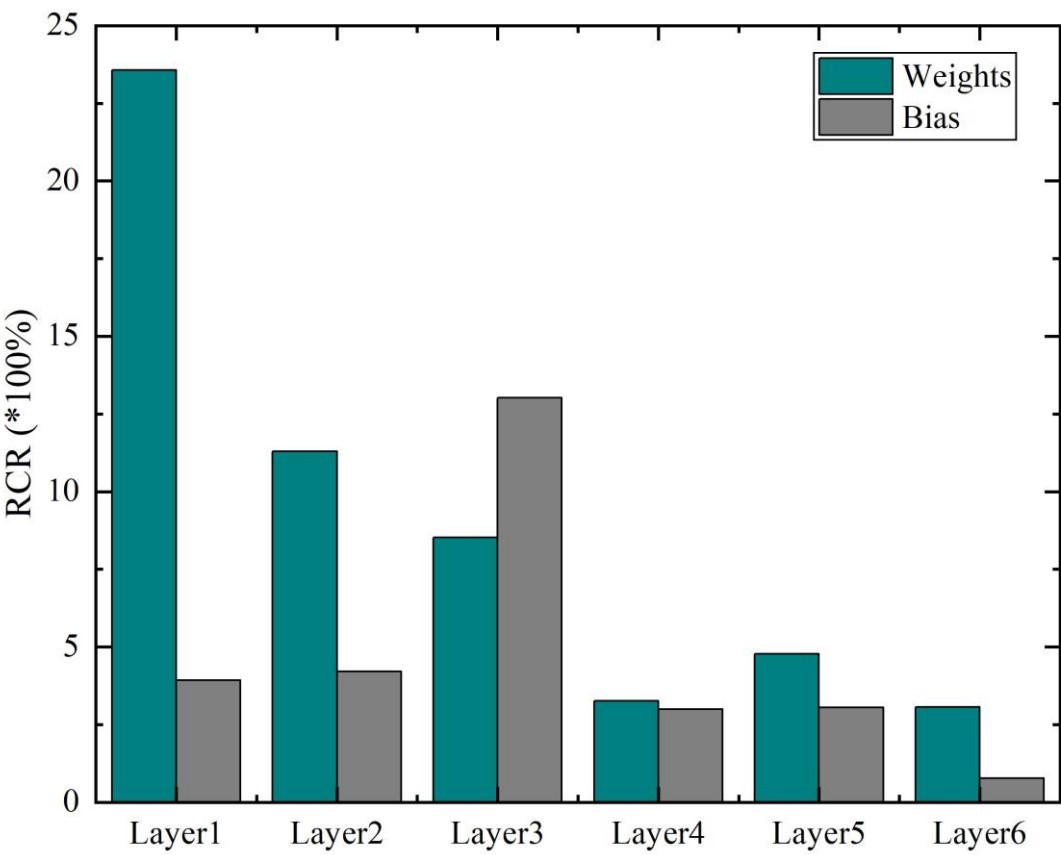

**Figure 4.** The average relative change rate ($RCR$) of weight between pre-training neural network with different $K$ values.





**Figure 5.** Absolute errors between the predicted temperature field and reference temperature field using

DTL and DNN models for homogeneous streambed.







**Figure 6.** Comparisons of the predicted temperature (blue curves) and reference temperature (red curves)

using DTL and DNN models for homogeneous streambed.





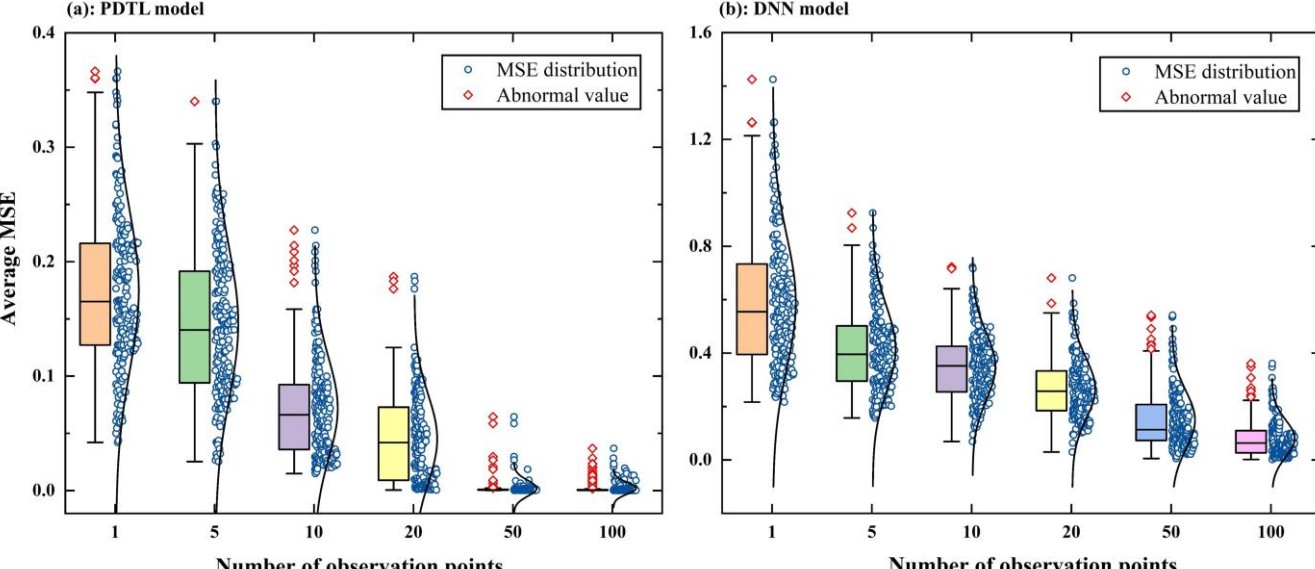

**Figure 7.** *MSE* distribution of normalized results from DTL and DNN models plotted against the number

of observation points for homogeneous streambed. (a) DTL model; (b) DNN model.



**Figure 8.** Heat map and the contours of hydraulic head and streamline for different *K*-fields. (a1) - (c1)

show the heat map and (a2) - (c2) show the contours of hydraulic head and streamlines for $\sigma_{lnk}^2 = 0.2$,

0.5, and 1.0, respectively.





**Figure 9.** *MSE* distribution of normalized results from DTL and DNN models plotted against the number

of observation points for different heterogeneous streambeds. (a1) - (c1) show the DTL model and (a2) -

(c2) show the DNN model for $\sigma_{lnk}^2 = 0.2$, 0.5, and 1.0, respectively.





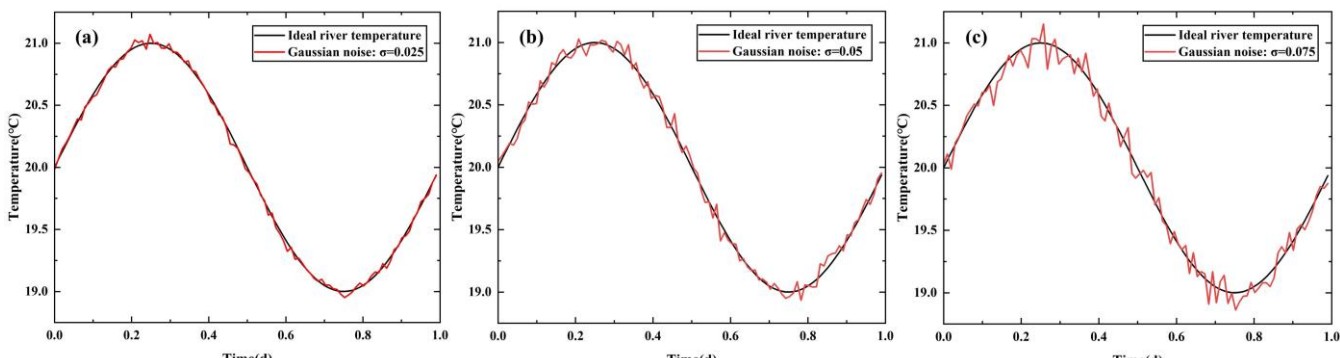

**Figure 10.** Time series diagram of river temperature under different observation noises. (a) $\sigma = 0.025℃$;

(b) $\sigma = 0.05℃$; (c) $\sigma = 0.075℃$.







510

**Figure 11.** *MSE* distribution of normalized results from DTL and DNN models plotted against the number of observation points for different observation noises. (a1) - (c1) show the DTL model and (a2) - (c2) show the DNN model for $\sigma = 0.025℃$, $0.05℃$, and $0.075℃$, respectively.