# Peer review of "Improving heat transfer predictions in heterogeneous riparian zones using transfer learning techniques"

_EGUsphere, 2024_

## Author Comment (AC1)

**CHINA UNIVERSITY OF GEOSCIENCES**

SCHOOL OF ENVIROMENTAL STUDIES

WUHAN, HUBEI, CHINA 430074

Dr. Quanrong Wang, Endowed CUG Scholar in Hydrogeology

Tel: +86 15927169156

Email: wangqr@cug.edu.cn

[Figure]

May 16, 2025
To: Dr. Heng Dai, Editor of Hydrology and Earth System Sciences
Subject: Revision of Paper # EGUSPHERE-2024-4145
**Dear Editor:**
Upon the recommendation, we have carefully revised Paper # EGUSPHERE-2024-4145 entitled "Improving
heat transfer predictions in heterogeneous riparian zones using transfer learning techniques" after
considering all the comments made by the reviewers. The following is the point-point response to all the
comments.
**Response to Reviewer #1:**

**Overview:**
This manuscript proposes a Deep Transfer Learning (DTL) approach to improve the accuracy of
spatiotemporal temperature distribution predictions in heterogeneous riparian zones. Using transfer learning,
the authors integrate analytical solution outputs for a homogeneous medium into a Deep Neural Network
(DNN) and employ a 2D numerical model output for a heterogeneous medium as their synthetic data. They
tested their approach by comparing the DTL to a DNN trained solely on synthetic data across various
heterogeneous media and noise levels. Their findings indicate that the DTL model outperforms the DNN
model in scenarios with limited training data and demonstrates greater robustness to data noise, which may
have practical applications in riparian zone management.
The current version of the manuscript requires significant work. Essential information regarding the physical-
based models used to train the DTL and DNNs is missing, as well as clarifications on the input and output
variables of the machine learning models needed for testing and reproducing the work presented. Additionally,
the authors should include the reasoning behind their sampling criteria and how it is linked to the physical
process they are modeling, as well as highlight how their novel framework differs or adds from work done by
previous authors. With the latter in mind, I cannot accept the manuscript in its current form.
Below, I have listed comments and suggestions, hoping they may help improve the manuscript's quality.

**Reply:** Thanks for your constructive comments. We have carefully revised Paper # EGUSPHERE-2024-4145.
**Specific Comments:**
**The physics-based models need further clarification.**
(1) The authors based their analytical and numerical models on previous work performed by Shi et al. (2023)
and they present some of the equations and boundary conditions in the manuscript and the supplementary
information. However, the manuscript does not clarify the actual domain of the system. Are they using the
model's domain as the conceptual model presented in Figure 1? If so, why are the modeling results presented
in a square? Is this an inset of the larger domain? If so, where is the inset located for the whole model? If it
is not an inset, is the domain different from the one presented by Shi et al. (2023)? If so, why is its extent shorter than that of the original study? A clear description of the conceptual model and its boundary conditions
should be included in the main manuscript to aid in the understanding of the physical process.

**Reply:** Implemented. Please see Lines 98-115 and section S1-"Analytical solution of the 2D heat transfer
process in homogeneous streambed" and S2-"Numerical solution of the 2D heat transfer process in
heterogeneous streambed" in the Supplement.
Figure 1 is the schematic of the conceptual model illustrating heat transfer induced by surface water-
groundwater interactions. Similar to Shi et al. (2023), the square domain used in this study represents a
simplified case of the conceptual model. The groundwater flow model and heat transfer model are coupled
through $q_x$ and $q_z$, which are directly defined in the analytical model used in this study. Therefore, only the
boundary conditions of the heat transfer model need to be considered in the analytical model. For
heterogeneous scenarios (numerical model), the boundary conditions of groundwater flow model are applied
to generate two-dimensional nonuniform flow fields, which are used to create fine-tuning and testing samples
under heterogeneous streambed conditions. The settings of initial and boundary conditions are shown in
Figure S2.
To improve simulation accuracy and avoid boundary effects in numerical model, the semi-infinite geometry
size was replaced by a finite range, and two infinite element domains were added at $x = 1$ m and $z = 1$ m
to represent the infinite boundary on the $x$- and $z$-directions, respectively. Therefore, the model's domain
size does not influence the accuracy of either the analytical or numerical solutions, and Figure S1 further
validates the accuracy of the proposed numerical model.

[Figure]

**Figure S1.** Comparison of temperature-time curves at three locations using the numerical solution (circle
symbols) and the analytical solution of this study (solid curves).

[Figure]

**Figure S2.** Conceptual model of 2D numerical model with streambed sediment, initial and boundary conditions. The red color zone represents the streambed with a half width of 0.32.

(2) Additionally, the groundwater flow model and its boundary conditions are not mentioned. Is this the same model as the one used in Shi et al. (2023)? This should be included and clarified in the manuscript for an integral understanding of the process that the data-driven models are trying to reproduce.

**Reply:** Implemented. Similar to Shi et al. (2023), the groundwater flow model and heat transfer model are coupled through $q_x$ and $q_z$. In the analytical model, $q_x$ and $q_z$ are directly prescribed; therefore, only the boundary conditions of the heat transfer model need to be considered in the analytical model.
For heterogeneous scenarios (numerical model), the boundary conditions of groundwater flow model are applied to generate two-dimensional nonuniform flow fields, which are used to create fine-tuning and testing samples under heterogeneous streamed conditions. The settings of initial and boundary conditions are shown in Figure S2.

(3) Incidentally, part of the work looks into heterogeneity, and the authors present their heterogeneous fields. Nonetheless, there is no mention of which hydraulic conductivity value is used for the homogeneous case. The authors only mention variations in the Darcy's fluxes ($q_x$ and $q_z$) in line 167. How are these fluxes calculated? What values are used for head gradients? Are the variations of these Darcy's fluxes related to boundary conditions or fluxes through the domain? I suggest including the Darcy flux equation and leaving the variations only to hydraulic conductivity to be consistent with the heterogeneous cases.

**Reply:** The groundwater flow model and heat transfer model are coupled through $q_x$ and $q_z$. In the analytical solution, $q_x$ and $q_z$ are directly prescribed. For the homogeneous case, we use two pairs of $q_x$ and $q_z$ values-(0.2 m/d, 0.3 m/d) and (0.6 m/d, 0.9 m/d)-to represent variations in hydraulic conductivity. The corresponding mathematical expressions are as follows:

$$q_x = -K_x \frac{\partial H}{\partial x} \tag{1}$$

$$q_z = -K_z \frac{\partial H}{\partial z} \tag{2}$$

$\quad v_x = \dfrac{C_w}{C_s} q_x = -K_x \dfrac{C_w}{C_s} \dfrac{\partial H}{\partial x}$ $\hspace{3cm}$ (3)

$\quad v_z = \dfrac{C_w}{C_s} q_z = -K_z \dfrac{C_w}{C_s} \dfrac{\partial H}{\partial z}$ $\hspace{3cm}$ (4)

$\quad C_s = (1 - \theta)\rho_s c_s + \theta \rho_w c_w$ $\hspace{3cm}$ (5)

where $H$ is the hydraulic head [L]; $q_x$ and $q_z$ are streambed water flux [LT−1] components of the streambed
on the $x$ and $z$-axes, respectively; $K_x$ and $K_z$ are the hydraulic conductivities [LT−1] on the $x$ and $z$-axes,
respectively; $v_x$ and $v_z$ are thermal front velocity [LT−1] components of the streambed on the x- and z-axes,
respectively; $C_w$ is specific volumetric heat capacity [J/(m3·°C)] of water; $C_s$ is specific volumetric heat
capacity [J/(m3·°C)] of streambed; $\rho_s$ and $\rho_w$ are densities [ML-3] of porous media and fluid, respectively;
$c_s$ and $c_w$ are specific heat capacities [J/(kg·°C)] of porous media and fluid, respectively. Please see section
S1-"Analytical solution of the 2D heat transfer process in homogeneous streambed" in the Supplement.
(4) The authors only present the fields for hydraulic conductivity and absolute errors, and there is no plot of
the temperature field they are trying to reproduce. Are these fields different from each other? How does the
heterogeneous domain affect the temperature distributions? I suggest adding a figure with the temperature
fields for the analytical and the numerical solutions so that the reader can understand how these fields vary
throughout the domain and what the data-driven models are missing.

**Reply:** Implemented. The transient temperature field is plotted in Figure 8. As mentioned in Section 3.1, the
transient temperature field consists of 100-time steps, resulting in 100 corresponding temperature field
figures. In this study, we selected the temperature field at $0.5d$ (i.e., the $50th$ time step) as the reference
field to calculate the absolute errors. We have added the reference temperature fields at $0.5d$ (i.e., the $50th$
time step) in Figure 8 to further demonstrated how heterogeneous hydraulic conductivity affect the
temperature distributions. Note that the selection of $0.5d$ is rather arbitrary for the demonstration purpose
and can be replaced by other time steps.
**With respect to the machine learning models**
(1) The authors mention in line 15 that this work "[proposes] a novel Deep Transfer Learning (DTL) approach
[…] to improve the accuracy of spatiotemporal temperature distribution predictions." However, a similar
approach has been explored in Zhang et al. (2023) for the prediction of hydraulic heads in heterogeneous
aquifers. The authors should clearly specify the improvements or modifications made to the framework
compared to Zhang et al. (2023), beyond the difference in application.

**Reply:** First, while Zhang et al. (2023) focused on one-dimensional groundwater flow, our study addresses
heat transfer in two-dimensional heterogeneous riparian zones. This involves fundamentally different
governing equations (convection-diffusion) than the groundwater flow equations in Zhang et al. (2023), more
complex physical processes (coupled flow and heat transport), and significantly higher computational
demands. Second, we incorporate physical constraints and impose penalties for violations of initial and
boundary conditions in the pre-trained DNN model by implementing an enhanced loss function. This
approach ensures that our model adheres to fundamental heat transfer principles. These innovations
collectively advance the application of deep transfer learning in environmental modeling beyond what was
presented in Zhang et al. (2023), with particular emphasis on heat transport processes in riparian zones and
a systematic evaluation of model robustness under various uncertainties. Please see Lines 133-154.

(2) In line 222, the authors mention that they restricted the number of epochs in the model training. Is there
a reason why these models cannot be trained with different epochs until they reach the same convergence?
Also, what about the other hyperparameters of the DNN models (i.e., number of nodes, number of layers,
epochs, and activation functions, among others), have the authors considered testing a range of these
parameters to get the best set of DNN?

**Reply:** We limited the number of epochs during model training primarily to reduce computational cost and
prevent overfitting. In addition, we employed an early stopping strategy to prevent overfitting by monitoring
the validation loss during training, thereby ensuring the model's generalization ability. This training strategy
is widely used in Wang et al. (2021), Zhang et al. (2023) and Wang et al. (2023), and their works have
demonstrated 3,000 epochs are sufficient for the DNN model to converge.
In addition, the other hyperparameters of the DNN models are selected after multiple trials. Results indicate
that the predictions of the two pre-trained models closely align with the analytical model, with average MSE
values of 1.2E-6 and 1.5E-6, respectively. This further demonstrates that the hyperparameters of the DNN
model are suitable.
Reference:
[1] Wang, N., Chang, H., and Zhang, D. (2021). Deep-learning-based inverse modeling approaches: A
subsurface flow example [J]. Journal of Geophysical Research: Solid Earth, 126, e2020JB020549.
[2] Zhang J, Liang X, Zeng L, et al. Deep transfer learning for groundwater flow in heterogeneous aquifers
using a simple analytical model [J]. Journal of Hydrology, 2023, 626: 130293.
[3] Wang, N., Chang, H., and Zhang, D. (2023). Inverse modeling for subsurface flow based on deep learning
surrogates and active learning strategies [J]. Water Resources Research, 59, e2022WR033644.
(3) Part of using these data-driven approaches is leveraging the current available data to predict variables
that are difficult, expensive, or impractical to measure. With this in mind, the authors should be clear about
what variables they are using as input to predict the temperature fields. Are they using the hydraulic heads
and temperature of the stream? Are they using variables related to the geology of the site? Or are they using
temperature data from previous timesteps? All of this is important because if we were to use these models
to predict the temperature in a given field site, we would need to know what variables we should measure to
be able to have an accurate prediction.

**Reply:** Implemented. This study focuses on improving heat transfer predictions in a heterogeneous
streambed using a deep transfer learning approach. We are concerned with the spatiotemporal thermal
distributions of the streambed. Therefore, we do not use previous hydraulic head, temperature, or geological
variables as input parameters. In this study, the input data consists of spatial locations $(x, y)$ and time $t$,
with dimensions of $100 \times 100 \times 100$. The output data is the corresponding temperature. We have clarified
that in the manuscript. Please see Lines 193-194.

(4) Furthermore, the authors should link their sampling criteria to the physical process they are trying to
reproduce with data-driven approaches. For instance, grabbing more than 50 samples in a 1-meter cross-
section with some spaced less than 0.1 meters horizontally is impractical and inefficient. I suggest the authors
approach the sampling criteria as they were placed in the field, and are tasked to maximize the location of
their thermistors or other measuring devices. This reviewer believes this approach can benefit the scientific
community and add value to the manuscript.

**Reply:** Implemented. We agree that linking our sampling approach to realistic field deployment scenarios
would enhance the manuscript's practical relevance.
In Section 3.3, our study deliberately evaluated various observation point densities (1, 5, 10, 20, 50, and 100)
to analyze the minimum monitoring requirements for effective model performance. Our results demonstrate
that the proposed PDTL model exhibits robust performance even with sparse data (fewer than 10 observation
points), which outperforms the traditional DNN approach under heterogeneous streambed conditions and
with observation noise. Please see Figures 5, 6, 7, 9, 11.
To establish statistically sound sampling criteria, we employed random sampling with 200 realizations for
each scenario, following established practices in field hydrology (Holmes et al., 2006; Ali et al., 2009). This
approach ensures unbiased sampling where every possible measurement location has an equal chance of
being selected, which is critical for comprehensive model evaluation. Similar sampling criteria have been
widely adopted in many recent studies (Goswami et al., 2022; Zhang et al., 2023).
Reference:
[1] Holmes, K. W., et al. (2006). Designs for marine remote sampling: a review and discussion of sampling
methods, layout, and scaling issues, Task 2.1 Milestone Report Published by the Cooperative Research
Centre for Coastal Zone, Estuary and Waterway Management (Coastal CRC).
[1] Ali, G.A. and Roy, A.G. (2009), Revisiting Hydrologic Sampling Strategies for an Accurate Assessment of
Hydrologic Connectivity in Humid Temperate Systems. Geography Compass, 3: 350-374.
[1] Goswami, S., Kontolati, K., Shields, M.D.et al. Deep transfer operator learning for partial differential
equations under conditional shift. Nature Machine Intelligence, 2022, 1155-1164.
[2] Zhang J, Liang X, Zeng L, et al. Deep transfer learning for groundwater flow in heterogeneous aquifers
using a simple analytical model [J]. Journal of Hydrology, 2023, 626: 130293.

(5) Consider including an additional paragraph or sentences that describe other approaches to create
physics-informed machine learning models (e.g., Arcomano et al., 2022; M. Raissi et al., 2019; Maziar Raissi
& Karniadakis, 2018; Yeung et al., 2022).

**Reply:** Implemented. We have added the paragraph to describe the physics-informed machine learning
models. Please see Lines 65-69.
Reference:
[1] Arcomano, T., Szunyogh, I., Wikner, A., Pathak, J., Hunt, B. R., & Ott, E. (2022). A hybrid approach to
atmospheric modeling that combines machine learning with a physics-based numerical model. Journal of
Advances in Modeling Earth Systems, 14(3), e2021MS002712.
[2] Raissi, M., Perdikaris, P., & Karniadakis, G. E. (2019). Physics-informed neural networks: A deep learning
framework for solving forward and inverse problems involving nonlinear partial differential equations. Journal
of Computational Physics, 378, 686–707.
[3] Jiang, S., Zheng, Y., and Solomatin, D (2020). Improving AI system awareness of geoscience knowledge:
Symbiotic integration of physical approaches and deep learning, Geophysical Research Letters, 47, 733-745.

[4] Kamrava, S., Sahimi, M., and Tahmasebi, P (2021). Simulating fluid flow in complex porous materials by
integrating the governing equations with deep-layered machines, npj Computational Materials, 7, 127.
[5] Arcomano, T., Szunyogh, I., Wikner, A., Pathak, J., Hunt, B. R., & Ott, E. (2022). A hybrid approach to
atmospheric modeling that combines machine learning with a physics-based numerical model. Journal of
Advances in Modeling Earth Systems, 14(3), e2021MS002712.
[6] Yeung, Y.-H., Barajas-Solano, D. A., & Tartakovsky, A. M. (2022). Physics-informed machine learning
method for large-scale data assimilation problems. Water Resources Research, 58(5), e2021WR031023.
[7] Zhao, W. L., Gentine, P., Reichstein, M., et al (2019). Physics-constrained machine learning of
evapotranspiration, Geophysical Research Letters, 46, 14496-14507.
[8] Cho, K. and Kim, Y.(2022). Improving streamflow prediction in the WRF-Hydro model with LSTM networks.
Journal of Hydrology, 605, 127297.

(6) I suggest the authors add more information in the discussion section. Where they highlight the importance
of their work and how it relates to other approaches. I suggest also highlighting the transferability of this
framework to other settings, as well as things that scientists should take into account.

**Reply:** Implemented. We have highlighted the transferability of our framework to other settings in the
discussion section. Please see Lines 308-322.

**Technical Corrections:**
(1) Lines 17-20 are difficult to read and contain variables that are not previously defined

**Reply:** Implemented. We have reorganized Lines 17-20 and removed the variables that are not previously
defined. Please see Line 18-24.

(2) The sentence in lines 22-23 is redundant, so consider removing it.

**Reply:** Implemented. We have removed Lines 22-23.

(3) Grammar in line 89 "Newly proposed demonstrates"

**Reply:** Implemented. We have revised the "newly proposed demonstrates" to "newly proposed approach
demonstrates". Please see Line 92.

(4) In line 294 should be "centers" instead of "centres"

**Reply:** Implemented. We have revised the "centres" to "centers". Please see Line 326.

(5) What do you mean by "it is postulated that the thermal and hydraulic properties of the streambed maintain
uniformity"? (Lines 98-99). Are you referring to the fact that these variables remain constant throughout the
simulation? Please clarify.

**Reply:** Implemented. It has been rephrased to provide better clarification. In fact, we want to express that
the thermal properties are homogeneous of the streambed. However, for hydraulic properties, i.e., hydraulic
conductivity, we considered both the homogeneous and heterogeneous scenarios. Please see Lines 101-
102.

(6) It should be "no heat flux boundary" in line 102.

**Reply:** Implemented. We have revised it to "no heat flux boundary". Please see Line 105.

(7) I recommend collapsing equations (1a) through (1c) to a single equation with a subscript i that is later
described.

**Reply:** Implemented. We have collapsed equations (1a) through (1c) to a single equation with a subscript $i$.
Please see equation (1a).

(8) Line 144 states that "The hyperparameters $\theta_T$ for the fine-tuning model is acquired through the
optimization of the loss function delineated by…" By definition, a hyperparameter cannot be estimated with
model training. They are set by the user. I think that you mean "The parameters" instead of "The
hyperparameters."

**Reply:** Implemented. We have revised the "hyperparameters $\theta_T$" to "parameters $\theta_T$". Please see Lines 170.

(9) Some variables, such as $q_x$ and $q_z$, are not defined in the main manuscript. Since the manuscript should
be self-contained, these variables should be specified in the text.

**Reply:** Implemented. We have defined $q_x$ and $q_z$ in the Supplement. Please see S1-"Analytical solution of
the 2D heat transfer process in homogeneous streambed" in the Supplement.

(10) Remember to add the units of the Mean Square Error (MSE) values.

**Reply:** To mitigate the impact of dimensionality during the training process, the temperature field dataset is
normalized to $[-1,1]$, and the temperature field dataset becomes dimensionless. Please see Lines 120-126.

(11) The text in Figures 5, 6, and 10 is difficult to read. Consider increasing the fonts. Also, include the units
of the variables plotted.

**Reply:** Implemented. We have increased the fonts and units of the variables in Figures 5, 6, and 10. Please
see Figures 5, 6, and 10.

(12) Consider using the same y-scale for Figures 7, 9, and 11. This would aid in the comparison.

**Reply:** Implemented. We have used the same y-scale for Figures 7, 9, and 11 for better comparison. Please
see Figure 7, 9, and 11.

**Response to Reviewer #2:**

**Overview:**

The manuscript presents a novel Deep Transfer Learning (DTL) framework for improving the prediction of spatiotemporal temperature fields in heterogeneous riparian zones. The authors leverage analytical solutions in homogeneous domains to pre-train a DNN model, and subsequently fine-tune it for heterogeneous cases, thereby addressing the challenge of limited observational data. The study is well-motivated and addresses an important problem in hydrological modeling. The methodology is clearly described, and the results are well-documented through a series of comprehensive experiments. I find the manuscript suitable for publication in Hydrology and Earth System Sciences after minor revisions. Below are my specific comments and suggestions for improving the manuscript.

**Reply:** Thanks for your constructive comments. We have carefully revised Paper # EGUSPHERE-2024-4145.

**General Comments:**

(1) While the manuscript briefly mentions physics-informed neural networks (PINNs), a more direct comparison or a deeper discussion of how DTL differs from or complements PINNs would strengthen the manuscript. This would better situate the DTL approach within the broader landscape of hybrid modeling techniques.

**Reply:** Implemented. In the revised manuscript, we also integrate multiple loss functions considers the constraints of physical information and imposes penalties for initial and boundary conditions for the pre-trained DNN model. Subsequently, the transfer learning technique is used to fine-tune the pre-trained model. Therefore, our model integrated the strength of PINNs and transfer learning technique .Please see Lines 133-154.

(2) The paper focuses on model performance but does not explore the interpretability of the DTL model. A short discussion on whether the transferred physical knowledge can be traced or interpreted in the model outputs would be beneficial. Furthermore, although the authors mention possible extensions to solute transport or other applications, this is not demonstrated or discussed in detail.

**Reply:** Implemented. In fact, we have discussed how the transferred physical knowledge impacts the parameters of deep learning model. Results indicate that the hydraulic conductivity primarily influences the parameters of the shallow layers in the DNN model. Please see Lines 196-202. Although a full interpretability analysis is beyond the current scope, we agree it is an important direction for future work and have noted this accordingly. Furthermore, we have discussed the extension of the proposed PDTL model to solute transport or other applications. Please see Lines 308-322.

(3) The authors acknowledge the limitation that analytical models assume regular geometries. This is an important point and could be expanded to discuss whether coordinate transformation, domain padding, or hybrid numerical-analytical datasets could mitigate this issue in future work.

**Reply:** Implemented. We have expanded this point in the discussion section. In future work, efforts should focus on improving the framework's ability to handle irregular spatial domains through coordinate transformations, domain padding, or hybrid numerical-analytical datasets, and on refining its extrapolation capability. Please see Lines 333-337.

**Specific Suggestions:**

(1) Section 2.2: Clarify why the tanh activation function is used rather than alternatives like ReLU. This choice may influence convergence and generalization.

**Reply:** Implemented. The tanh activation function was selected primarily due to its bounded and symmetric output range (-1 to 1), which helps in centering the data and can lead to faster convergence during training, especially when the inputs have been normalized. In our case, since the model learns from inputs that include both positive and negative physical feature, $tanh$ facilitates smoother gradient flow across layers. Furthermore, the tanh activation function is widely used in hydrological problems, e.g., An et al. (2022), Zhang et al. (2023).

References:
[1] An, Y., Yan, X., Lu, W. et al. An improved Bayesian approach linked to a surrogate model for identifying groundwater pollution sources. Hydrogeology Journal, 2022, 601-616.
[2] Zhang J, Liang X, Zeng L, et al. Deep transfer learning for groundwater flow in heterogeneous aquifers using a simple analytical model [J]. Journal of Hydrology, 2023, 626: 130293.

(2) Equation (2): Notation should be consistent with Equation (4). Clarify the definition of n (number of training samples).

**Reply:** Implemented. We have incorporated the collocation points throughout the model domain, along with the physical information of the boundary and initial conditions to minimize the loss functions. Please see Equations (3a)-(3d).

(3) Consider including results for 200 observation points in the main figures, rather than relegating them to the Supplement, since these are discussed prominently in the text.

**Reply:** Considering data acquisition in heterogeneous riparian zones is often time-consuming and costly, this study focuses on the performance of the proposed PDTL model under limited data availability. There is no significant difference between the PDTL and DNN model when the number of observation points increases to 200, therefore we relegate them to the Supplement.

If you have any further questions about this revision, please contact me.
Sincerely Yours,
Quanrong Wang, PhD, PG.
Professor and
Holder of Endowed CUG Scholar in Hydrogeology

---

## Author Response (AR2)

**CHINA UNIVERSITY OF GEOSCIENCES**

**SCHOOL OF ENVIROMENTAL STUDIES WUHAN, HUBEI, CHINA 430074**

Dr. Quanrong Wang, Endowed CUG Scholar in Hydrogeology Tel: +86 15927169156 Email: wangqr@cug.edu.cn

July 18, 2025

To: Dr. Heng Dai, Editor of Hydrology and Earth System Sciences

Subject: Revision of Paper # EGUSPHERE-2024-4145

5

**Dear Editor:**

Upon the recommendation, we have carefully revised Paper # EGUSPHERE-2024-4145 entitled "Improving heat transfer predictions in heterogeneous riparian zones using transfer learning techniques" after considering all the comments made by the reviewers. The following is the point-point response to all the comments.

11

**Response to Reviewer #1:**

**12 Overview:**

- Thank you for submitting your manuscript addressing the comments previously made. This new version of the manuscript is in better shape. I only have a few comments and one technical correction that are presented
- 15 below.
- 16 Reply: Thanks for your constructive comments. We have carefully revised Paper # EGUSPHERE-2024-4145.

18

**Specific Comments:**

- 19 (1) It is still not clear how the authors find the prescribed values for Darcy's fluxes  $(q_x \text{ and } q_z)$  in the analytical solution. The authors argue in their response (lines 91 to 96 in the response document) that the groundwater
- flow and heat transfer models are coupled through  $q_x$  and  $q_z$ , which they prescribed in the analytical solution.
- The numerical model has two boundary conditions where a constant head is set (Figure S2). Did the authors
- 23 calculate the Darcy's fluxes using the difference between these heads throughout the domain and the given
- 24 hydraulic conductivity, or did they use the Darcy's flux simulated results for a given cell within the domain? I
- 25 recommend providing further clarification on this.
- Reply: Implemented. In the analytical model of Shi et al. (2023), the thermal front velocities  $v_x$  and  $v_z$  are
- directly prescribed, please see Eqs.(1)-(10).

$$T = A(x,z,t) \left[ \int_0^\infty \int_o^L E(\lambda,\eta) B(x-\lambda,z-\eta,t) d\lambda d\eta + \int_o^t g(\tau) C(x,z,t-\tau) d\tau \right]$$
 (1)

$$A(x,z,t) = \frac{1}{4} exp \left[ -\frac{v_z(v_z t - 2z)}{4D_z} - \frac{v_x(v_x t - 2x)}{4D_x} \right]$$
 (2)

$$B(x-\lambda,z-\eta,t) = \frac{1}{\pi t \sqrt{D_x D_z}} \left\{ exp \left[ -\frac{(z-\eta)^2}{4D_z t} \right] - exp \left[ -\frac{(z-\eta)^2}{4D_z t} \right] \right\}$$

$$\quad \left\{ exp\left[ -\frac{(x-\lambda)^2}{4D_x t} \right] + exp\left[ -\frac{(x-\lambda)^2}{4D_x t} \right] \right\} \tag{3}$$

$$C(x,z,t-\tau) = \frac{z}{\sqrt{\pi D_z(t-\tau)^3}} exp\left[-\frac{z^2}{4D_z(t-\tau)}\right] \left\{ erf\left[\frac{L-x}{\sqrt{4D_x(t-\tau)}}\right] + erf\left[\frac{L+x}{\sqrt{4D_x(t-\tau)}}\right] \right\}$$
(4)

$$E(\lambda, \eta) = h(\lambda, \eta) exp \left[ -\frac{2\eta v_z}{4D_z} - \frac{2\lambda v_x}{4D_x} \right]$$
 (5)

$$g(\tau) = f(\tau)exp\left[\frac{v_z(v_z\tau)}{4D_z} + \frac{v_x(v_x\tau - 2x)}{4D_x}\right]$$
 (6)

$$q_x = -K_x \frac{\partial H}{\partial x} \tag{7}$$

$$q_x = -K_x \frac{\partial H}{\partial x}$$
 (7)
$q_z = -K_z \frac{\partial H}{\partial z}$  (8)

$$v_{z} = \frac{c_{w}}{c_{c}} q_{z} = -K_{z} \frac{c_{w}}{c_{c}} \frac{\partial H}{\partial z}$$
 (10)

- One can find that  $v_x$  and  $v_z$  are directly prescribed in the analytical model, which can be transformed into 39 the Darcy's fluxes through Eqs.(9)-(10). 40
- 41 For heterogeneous scenarios (numerical model), the boundary conditions of groundwater flow model are 42 applied to generate two-dimensional nonuniform flow fields. The Darcy's fluxes are calculated using the 43 difference between these heads throughout the domain The settings of initial and boundary conditions are 44 shown in Figure S2.
  - (2) The main objective of the manuscript is to propose a novel physics-informed deep transfer learning (PDTL) approach to improve the accuracy of spatiotemporal temperature distribution predictions. The manuscript, as it is, works towards that goal and presents a promising methodology. My only concern is the use of the location in the domain as the only input variable for the machine learning model. This limits the transferability of the model to other potential locations and disconnects the association of the model to measurable physical variables, such as the temperature at the surface and the fluxes of water from the river to the underlying aguifer. Some of these limitations are already presented in the discussion, but I encourage the authors also to discuss the limitations of their selected input variables.
  - Reply: We agree that using locations as inputs may limit the model's transferability to other sites and weaken its direct connection to measurable physical variables. We have added further discussion on this limitation in the revised manuscript. Future work will incorporate additional physically measurable parameters, such as surface temperature, river-aguifer fluxes, or hydraulic gradients, to enhance the model's generalizability and physical relevance. Please see Lines 335-339.

**Technical Correction:**

- (1). The text in the axes of Figures 5, 6, and 9 is difficult to read. Consider increasing the fonts. Also, include the units of the variables plotted.
- 63 Reply: Implemented. We have increased the fonts and added the units of the variables in Figures 5, 6, and 64 9.

**Response to Reviewer #3:**

**Overview: 66**

46

50

60

62

Thank you for the opportunity to review this thoroughly revised manuscript. The authors present a novel hybrid framework that combines simple analytical solutions for homogeneous systems with transfer learning 68 69 techniques to address heterogeneous heat transfer problems in riparian zones. The topic is timely and of 70 broad interest, and the methodology, which leverages well understood analytical models to inform data driven 71 learning, represents a creative and potentially widely applicable approach. The revised manuscript is significantly improved in organization, clarity, and technical depth compared to the previous version, and I recommend acceptance after a few clarifications and minor edits.

**Reply:** Thanks for your constructive comments. We have carefully revised Paper # EGUSPHERE-2024-4145.

**General Comments:**

(1) First, the rationale for using both analytical and numerical models to generate training data and benchmarks would benefit from a more explicit explanation. Although the hybrid strategy intuitively combines the interpretability and low-cost evaluation of analytical solutions with the flexibility of numerical simulations, the manuscript would be stronger if it contrasted this approach against purely numerical or purely statistical alternatives. For example, how does the introduction of analytical physics reduce the required volume of high-fidelity simulations and how does it improve generalization to untested heterogeneous conditions? A brief discussion of these tradeoffs and any computational savings or improved convergence properties observed would help readers appreciate the advantages of the proposed framework.

**Reply:** Implemented. We have added explicit rationale for employing both analytical and numerical models to generate training data and benchmarks. Please see Lines 171-176.

In this study, we focus on the physical principles transferred from homogeneous to heterogeneous streambed using transfer learning techniques. Accordingly, the source and target datasets are spatiotemporal distributions of temperature fields in homogeneous and heterogeneous streambeds, respectively. The analytical model developed by Shi et al. (2023) is employed to provide training dataset for pre-training. However, for the heterogeneous streambed, the analytical model is not available; therefore, numerical models are employed to generate the fine-tuning dataset and serve as the benchmarks to evaluate the performance of the proposed PDTL model.

We also conducted the comparative discussion between the hybrid approach (PDTL model) and or pure data-driven method (DNN model). Results indicate that the PDTL model significantly outperforms the DNN model in scenarios with scarce training data. Furthermore, increasing streambed heterogeneity and observation noise levels raises the mean MSE values of the PDTL and DNN models, with the PDTL model exhibiting greater robustness than the DNN model, highlighting its potential for practical applications in riparian zone management. Please see Section 3.2-3.4.

(2) Second, the choice of Shi et al. (2023) as the benchmark analytical model deserves further justification. The literature contains numerous analytical expressions for subsurface heat transport in riparian contexts; explaining why this formulation was selected (for example, because of its balance of simplicity and fidelity, its treatment of boundary conditions, or its previous validation against field data) would clarify its role in the study. If other models were considered but found less suitable, a sentence or two outlining those comparisons would reinforce confidence in the benchmark's relevance.

**Reply:** Implemented. We have added a justification for selecting the analytical model of Shi et al. (2023) as the benchmark in our revised manuscript. Specifically, we clarified that this model offers a balanced trade-off between analytical simplicity and accuracy, particularly in representing boundary conditions relevant to heat transport between surface water and groundwater. Moreover, this model has been validated against field data, further enhancing its credibility. Please see S1-"Analytical solution of the 2D heat transfer process in homogeneous streambed" in the Supplement.

(3) Third, the description of the transfer learning workflow indicates that some network layers were frozen while others remained trainable, but the manuscript does not specify which layers were selected nor the criteria guiding these decisions. Since layer freezing can critically affect the retention of low-level physical features versus high level adaptation to heterogeneous data, please detail which layers were frozen, which were fine-tuned, and why. Were early convolutional filters preserved to encode general diffusion patterns while later dense layers were adapted to capture heterogeneity? Clarifying this architecture will help readers reproduce the experiments and understand how transfer learning choices influence model performance.

**Reply:** Implemented. The difference in weights and biases between the two pre-training models with different hydraulic conductivities is evaluated using the relative change rate (*RCR*):

$$RCR = \frac{1}{I} \sum_{i}^{I} \frac{|\theta_{1i} - \theta_{2i}|}{\theta_{1i}} \tag{11}$$

where  $\theta_{1i}$  and  $\theta_{2i}$  are parameter matrixes in two pre-training models, respectively, I is the number of elements in the parameter matrix. For enhanced comparability and credibility, each of the two pre-training models undergoes 20 training processes. Figure 1 presents the average RCR of weights and biases across all layers for two per-training models over 20 trials. The RCR of biases shows consistent stability across all layers, except for layer 3. In contrast, variations in weights are more prominent, particularly in layers 1, 2, and 3, which underscores the heightened sensitivity of these layers to hydraulic conductivity. Consequently, layers 1, 2, and 3 of the pre-training models are marked as trainable, while the remaining layers are held frozen in the following analysis. Please see Section 3.1.

**Figure 1.** The average relative change rate (RCR) of weight between pre-training neural network with different K values.

(4) Fourth, the results demonstrate that transfer learning informed by analytical solutions performs remarkably well, even in heterogeneous systems that deviate from the homogeneous assumptions underlying the analytical form. It would be valuable to discuss potential reasons for this robustness: for instance, do the analytical solutions capture the dominant modes of heat propagation that persist under moderate heterogeneity? Is there a particular physical principle or scaling law embedded in the base model that remains valid across a range of conductivity contrasts? A short analysis of what features the network retains from the analytical initialization and how those guide learning in more complex settings would deepen insight into why the approach succeeds.

**Reply:** Implemented. The proposed PDTL model integrates advantages of analytical solutions, deep learning models, and transfer learning techniques. The analytical model is used to efficiently produce physically consistent heat distribution patterns and data in homogeneous riparian zones, which serve as the training data for the pre-training deep learning model. Subsequently, the weights and biases learned from the pre-training model are transferred to a new deep learning model under heterogeneous scenarios through transfer learning. Furthermore, we have discussed how the transferred physical knowledge impacts the parameters of deep learning model. Results indicate that the hydraulic conductivity primarily influences the parameters

- of the shallow layers (Layers 1-3) in the DNN model. Therefore, layers 1, 2, and 3 of the pre-training models are marked as trainable, while the remaining layers are held frozen in the following analysis. Please see
- 154 Lines 192-215.

- 156 Specific Suggestions:
- 157 (1) In Lines 65-73 "DL" should be "deep learning"
- 158 **Reply:** Implemented. We have revised "DL" to "deep learning". Please see Lines 65-73.

- 160 (2) In Line 87, "transfer learning" should be "transfer learning techniques".
- 161 **Reply:** Implemented. We have revised "transfer learning" to "deep learning techniques". Please see Line 87.

- 163 (3) In Line 91, "power" should be "capability".
- Reply: Implemented. We have revised "power" to "capability". Please see Line 92.

- 166 (4) In Line 106, L=0.32m is used in Section 2.1 or in all cases?
- 167 **Reply:** Implemented. L = 0.32m is used in all cases.

- 169 (5) In Lines 138, 151 and 152, "physical information" should be "physical principles".
- 170 Reply: Implemented. We have revised "physical information" to "physical principles". Please see Lines 138-152.

- 173 (6) In Lines 207-208, add references to support this point.
- Reply: Implemented. We have added some references to support our point to restrict the number of epochs in the model training process.

- 177 Reference:
- [1] Wang, N., Chang, H., and Zhang, D. (2021). Deep-learning-based inverse modeling approaches: A subsurface flow example [J]. Journal of Geophysical Research: Solid Earth, 126, e2020JB020549.
- [2] Zhang J, Liang X, Zeng L, et al. Deep transfer learning for groundwater flow in heterogeneous aquifers using a simple analytical model [J]. Journal of Hydrology, 2023, 626: 130293.

Quarong Wang

- 184 If you have any further questions about this revision, please contact me.
- 185 Sincerely Yours,
- 186 Quanrong Wang, PhD, PG.
- 187 Professor and
- 188 Holder of Endowed CUG Scholar in Hydrogeology